# Development of a Decision Matrix for National Weather Service Red Flag Warnings

**Sarah Jakober [1], Timothy Brown [2,\*] and Tamara Wall [2]**

[1] USFS Wallowa Whitman National Forest, La Grande, OR 97850, USA; sarah.jakober@usda.gov
[2] Desert Research Institute, Reno, NV 89512, USA; tamara.wall@dri.edu
\* Correspondence: tim.brown@dri.edu

**Abstract:** The National Weather Service is responsible for alerting wildland fire management of meteorological conditions that create an environment conducive for extreme fire behavior. This is communicated via Red Flag Warnings (RFWs), which presently lack a national standardized methodology and rarely are explicitly linked to fuel conditions such those as provided by National Fire-Danger Rating System (NFDRS) indicators. The need for a revamped RFW has been expressed recently by both fire management and fire weather meteorologists. A decision matrix approach was developed to determine criteria that consistently and explicitly associates meteorological and fuels information to extreme fire behavior. Extreme fire behavior is defined here as maximum rates of spread (area per day) observed on documented large fires from 1999–2014 utilizing the ICS209 all-hazard dataset. Meteorological conditions occurring with these rates of spread were compared to historical percentiles of relative humidity, wind speed, and the NFDRS Energy Release Component. These percentiles were assigned a numerical score from one through five based on percentile rank. The additive result of all three scores was plotted against rates of spread yielding a two-step decision matrix of RFW categories where, for example, the highest score is the most extreme RFW case. Actual RFW issuances were compared to this matrix method.

**Keywords:** Red Flag Warning; extreme fire behavior; National Fire-Danger Rating System; fire weather

## 1. Introduction

Wildfire is both a natural and anthropogenic impact on the landscape that presents an increasing hazard to society [1]. The increasing presence and severity of wildfire [2–4] will demand more accurate prediction methods, better community preparedness, and the increased use of early warning systems to mitigate the negative impacts of increasing burned area [5], seasonal duration [6], and more extreme fire behavior. In the United States, National Oceanic and Atmospheric Administration (NOAA) National Weather Service (NWS) meteorologists issue Fire Weather Watches (FWWs) and Red Flag Warnings (RFWs) to alert land managers when a "combination of fuel and weather conditions support extreme fire danger and/or fire behavior" [7] (p. 6). FWWs are generally issued up to 72 h in advance of forecasted conditions, while RFWs are issued within 48 h. Meteorological criteria for RFWs are selected by each weather forecast office (WFO) and documented in an interagency Annual Operating Plan [7]. RFWs are issued for a particular fire weather zone (FWZ) and are commonly based on predetermined thresholds of wind speed and relative humidity. RFWs are also required to incorporate fuels information, making them one of the few products issued by the NWS that utilize external non-NWS derived information [8]. Despite this requirement, explicit fuels information is not consistently used at all WFOs.

RFWs are used operationally by fire managers for planning and safety purposes [7]. Resource allocation and staffing decisions are common planning actions that may be based in part on a RFW as outlined in an agency fire danger operating plan (e.g., https://gacc.nifc.gov/nwcc/districts/CCCC/nfdrs2016/docs/WW_FDOP_final_draft_v20200428.pdf accessed on 18 March 2023). Essential safety components of operational firefighting tactics,

training, and practices involve the awareness or avoidance of extreme fire behavior [9]. Since RFWs are intended to alert fire managers of the potential for extreme fire behavior, they are an essential safety tool. It has previously been hypothesized that RFWs may have a measurable effect on firefighter situational awareness, potentially affecting the outcome of suppression efforts [10].

Very little research exists on the verification of FWWs and RFWs. Recent work published by Clark [10] demonstrated the skill of RFWs as forecasts of large fire occurrences, making several recommendations including the adoption of a standardized method of incorporating fuel dryness, centrally documented and explicit criteria, and the addition of probabilistic information. Several issues with RFWs have previously been identified [11]: current criteria are subjectively determined and are not scientifically linked to extreme fire behavior, fuels information is inconsistently incorporated, and the current warning structure limits a forecaster's ability to communicate uncertainty or relative severity. During a preliminary investigation done at the Desert Research Institute, at least 524 unique criteria were discovered from 40 separate Annual Operating Plans across the country [12], clearly demonstrating the need for a standardized methodology. The aim of the present research is to propose a national standardized methodology utilizing commonly observed meteorological elements of relative humidity and wind speed and incorporating an indicator of fuel dryness. By attempting to align the criteria with documented incidents of extreme fire behavior, RFWs will more effectively communicate information to fire managers, and subsequently improve planning and resource allocation, and increase firefighter and public safety as intended.

Scientifically, there is currently no formally accepted definition of extreme fire behavior [13]. The National Wildfire Coordinating Group (NWCG) defines extreme fire behavior as "a level of fire behavior characteristics that ordinarily precludes methods of direct control action . . . Predictability is difficult because such fires often exercise some degree of influence on their environment and behave erratically, sometimes dangerously" [14]. The preclusion of direct control action may result from several context-dependent and interrelated factors, summarized here by Tedim as the "interplay among macro processes (e.g., atmosphere and fire interaction) and local processes and conditions (e.g., poor initial attack, inadequate risk perception, very strong and variable winds, rough topography, low fuel moisture content, fuel load, fuel continuity, landscape connectivity, poor preparedness, and vulnerable communities)", [13] (p. 20).

Due to this complexity, fire behavior is commonly correlated exclusively to atmospheric processes. These correlations have been demonstrated using components from the US National Fire Danger Rating System (NFDRS), as well as other meteorological information [15–20]. Besides operational NFDRS there are other fire danger indicators that incorporate NFDRS elements [19,21] and others have been proposed that focus strictly on atmospheric conditions [22,23].

The documented instances of extreme fire behavior used in this project are taken from the ICS-209 all-hazard dataset, mined from the United States National Incident Management System [24], which consists of all large wildfire incidents in the United States from 1999–2014 recorded with an ICS-209 form [25]. By choosing this dataset, we are effectively controlling for incidents that exceeded initial attack capabilities, often implying some form of problematic fire behavior. Although these incidents may not have explicitly 'precluded direct control action' [14], it is highly likely that atmospheric conditions allowed for rapid initial spread. Only 1–2% of all wildfires in the US become large incidents, though these incidents account for over 95% of annual area burned [26].

Here, we examined the meteorological and environmental conditions surrounding documented occurrences of high rates of spread using relative humidity (RH), wind speed, and energy release component (ERC; an indicator of fuel dryness and total heat release per unit area within the flaming front at the head of a moving fire), parameters previously demonstrated to correlate to increased fire behavior [16,27,28]. Lindley [29,30] also shows a correlation between ERC and various measures of extreme fire behavior. Nomograms

depicting increases in rate of spread and flame length with corresponding increases of fire intensity and decreases in dead fuel moisture directly relate ERC and RH to fire behavior [31]. Wind is a variable in the original rate of spread equation proposed by Rothermel [32]. ERC, RH, and wind speed are used as inputs in the fire behavior model currently generating NFDRS fire danger ratings (Figure 1).

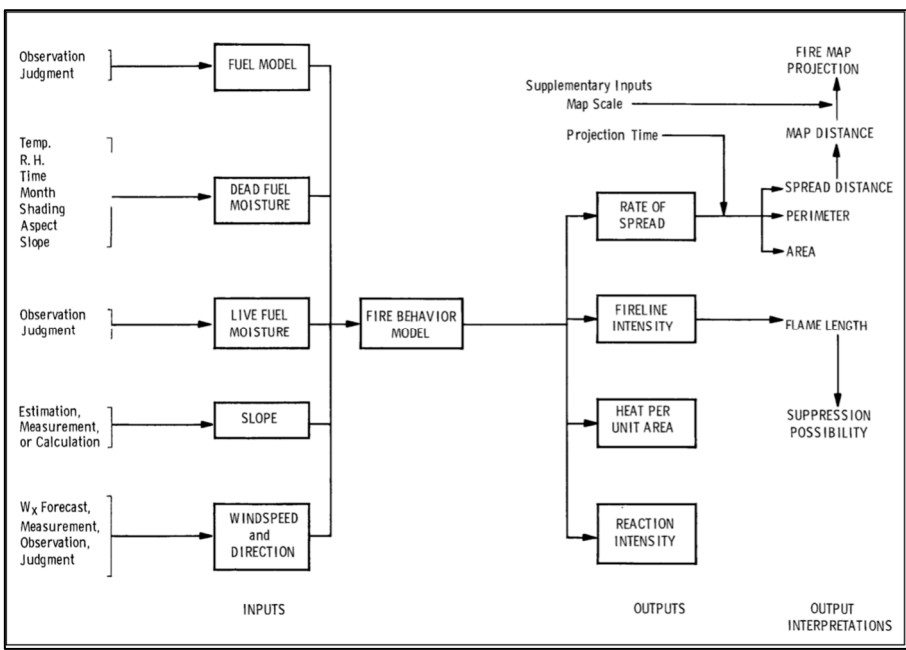

**Figure 1.** Inputs for the NFDRS fire behavior model. Note the presence of RH, live and dead fuel moistures (ERC; RH), and wind speed. With the exception of temperature, all other variables included in the NFDRS model are either site-specific or time related. Adapted from How to Predict the Spread and Intensity of Forest and Range Fires (INT-143) [33] (p. 2), Intermountain Forest and Range Experiment Station: Ogden, UT: USDA Forest Service. Retrieved from https://www.fs.fed.us/rm/pubs_int/int_gtr143 accessed 18 March 2023. In the public domain.

It is relevant to incorporate ERC along with existing criteria of wind speed and RH for several reasons: current RFW criteria inconsistently and non-uniformly incorporate fuel dryness, ERC can be normalized relative to a local climatology and correlate fuels information exclusively to meteorological conditions [19], and fire danger indices are operationally understood by fire management, the intended audience of RFWs [34]. By formulating RFW criteria using historical percentiles of these parameters we capture the dominant meteorological influences on fire behavior and directly relate RFWs to the potential for extreme fire behavior.

The goal of our work is to quantify the relationship between meteorological conditions and extreme fire behavior, offering a national standardized methodology for incorporation into operational fire weather forecasting. Better aligning weather and fuels with fire behavior will allow fire managers to be better informed of the likelihood and severity of a forecasted fire weather event with potential implications for firefighter safety and fire suppression resource management. A well-defined early-warning system with relevant communication messaging has the potential to increase occupational safety [35], better warn fire-threatened communities [36], and allow for more accurate tactical planning [37].

## 2. Materials and Methods

### 2.1. Fire Behavior and Gridded Meteorological Datasets

To analyze occurrences of extreme fire behavior, a database of documented large fires occurring in the United States from 1999–2014 containing the daily maximum rate of spread corresponding to each incident was acquired [24]. A large fire is defined here as anything

exceeding 100 acres (247 hectares) in a timber fuel type and 300 acres (741 hectares) in grass or brush fuel types. These maximum rates of spread were originally listed as area (acres) of growth per day with the calendar date at which the maximum occurred, documenting the most active fire behavior occurrence for each incident. Maximum rates of spread, dates of occurrence, and geographic locations were collected from twenty states in diverse geographic regions (see Table 1). By only examining definitively large fires, we effectively controlled for increased or above average fire behavior as an incident must escape initial attack and progress to certain acreage requirements to qualify for the database [24].

**Table 1.** Data sampled by state, including ICS209 incidents exceeding breakpoint rates of spread, total 2020 RFWs issued, and total 2020 RFWs corresponding to a FWZ with a RAWS station or a previous large fire occurrence.

| State | ICS209 Qualifying Incidents | ROS Threshold (Hectares) | 2020 Total RFWs Issued | 2020 RFWs Sampled |
|---|---|---|---|---|
| Alabama | 16 | 1255 | 4 | 4 |
| Arkansas | 21 | 1517 | 5 | 5 |
| Arizona | 27 | 21,745 | 149 | 149 |
| California | 64 | 21,498 | 643 | 566 |
| Colorado | 28 | 14,826 | 1143 | 603 |
| Florida | 100 | 5090 | 126 | 76 |
| Idaho | 49 | 32,297 | 138 | 138 |
| Kentucky | 11 | 2224 | 10 | 10 |
| Minnesota | 12 | 13,220 | 185 | 42 |
| Missouri | 10 | 1633 | 434 | 114 |
| Montana | 38 | 27,676 | 327 | 327 |
| N. Carolina | 14 | 3212 | 108 | 75 |
| Nebraska | 7 | 25,328 | 75 | 75 |
| Nevada | 44 | 42,626 | 315 | 267 |
| Oklahoma | 106 | 4942 | 199 | 117 |
| Oregon | 24 | 49,564 | 176 | 176 |
| Texas | 41 | 39,537 | 668 | 418 |
| Utah | 40 | 20,235 | 397 | 374 |
| Virginia | 11 | 2595 | 9 | 6 |
| Washington | 21 | 28,714 | 84 | 80 |
| Totals | 684 | Avg. 17,987 | 5195 | 3622 |

RFWs and FWWs are exclusively intended for conditions conducive to extreme fire behavior. Consequently, the focus of this analysis was to categorically demonstrate the severity of the most destructive large fires using a matrix methodology still applicable in daily operational forecasting. Maximum rates of spread were separated by state and graphed in ascending order, yielding the relationships demonstrated in Figure 2. Here, we further excluded all but the most extreme examples of fire behavior by selecting only those incidents occurring on the right side of a breakpoint, plotted to signify the point at which rates of spread start rapidly increasing. These breakpoints were obtained using an analytical method developed in Python specifically designed to find the knee or elbow of curves by calculating the slope of a line drawn from the minimum to maximum value, rotating the data so that the slope of this line is zero, and then finding the minimum value of the rotated data, corresponding the inflection point of the curve [38].

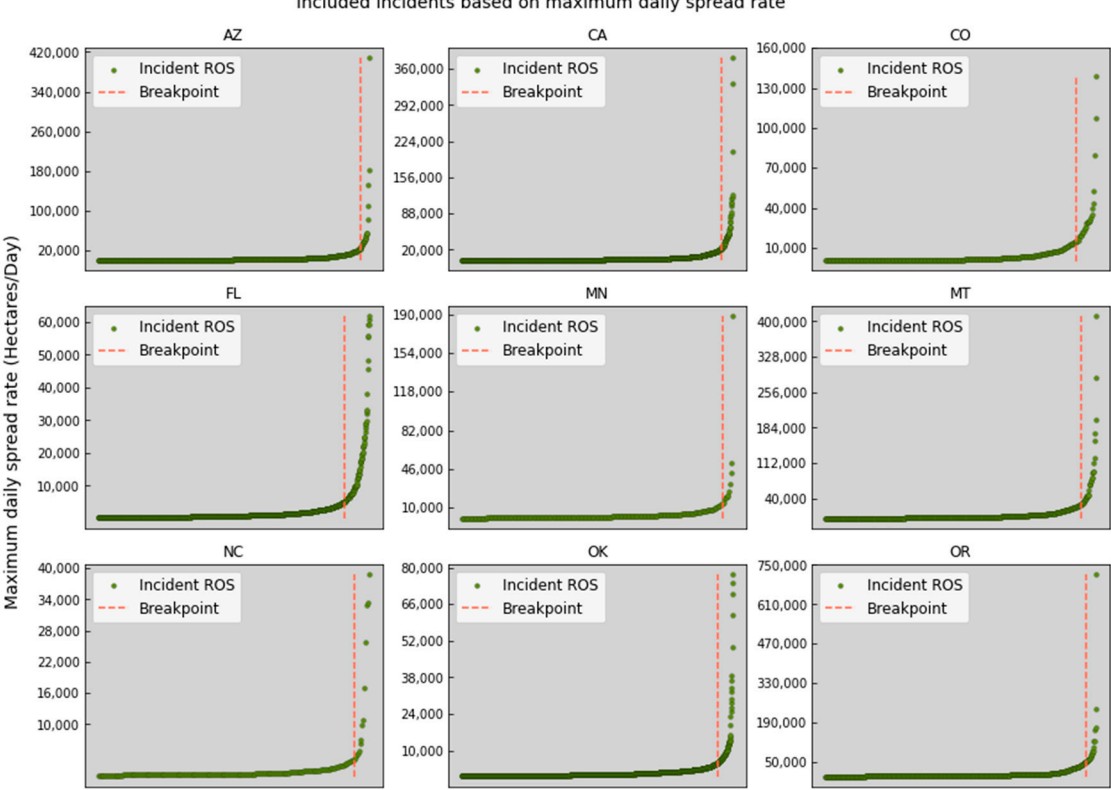

**Figure 2.** Daily maximum spread rate for incidents occurring in AZ, CA, CO, FL, MN, MT, NC, OK, and OR, 1999–2014, plotted in ascending order. ROS breakpoint is graphically demonstrated by the vertical dashed line. Incidents retrained for analysis are plotted to the right of the breakpoint while excluded incidents are plotted to the left.

To compile meteorological conditions corresponding to each incident, a 4-km gridded surface climatology was sampled [39], containing energy release component (ERC-G in the legacy NFDRS systems, ERC-Y in the new NFDRSv4); BTU/ft2), daily minimum relative humidity (%), and wind speed (mph) measured at 1300 LST. Although ERC-G refers to ERC calculated for that specific fuel type (fuel model G, conifer forest), it is considered conventional to standardize the fuel model when evaluating the index across large geographical areas [19,40]. This reduces any variability that may exist between fuel types and allows analyses to focus on relative meteorological change over time.

*2.2. Scoring Historical Percentiles*

ERC, RH, and wind speed values obtained from the gridded climatology for the date of each qualifying incident were normalized to site-specific values spanning 1999–2014 to yield percentiles. These percentiles were assigned a score according to the categories listed in Table 2. These categories are arbitrary, and were chosen to facilitate systematic, cumulative scoring when applied to a decision matrix. Of all the large fires included in this analysis, very few occurred when the ERC was below the 50th percentile and RH above the 50th percentile, thus categorical division began at these percentiles to better tune matrix scoring ability. The sum of each category represented the total score for the incident, 15 being the most extreme. By individually scoring each parameter, we provided a method of quantifying the total severity of a given set of meteorological conditions. The alignment of high scores in all three categories represent the most extreme conditions favorable to large rates of spread [33]. This method allows for the total score to reflect the severity of each individual category. For example, if RH and ERC both score a 5 and wind speed scores a 2, the total score is still relatively high (12). Conversely, if ERC and RH are both moderate at 3 and 3 and wind speed is high at 4, the total score is 10. Similar methods

are currently being used operationally to score wind speed and RH criteria against each other by some WFOs [41]. The datasets used in this work did not reveal any obvious mathematical thresholds, and the behavior of large wildfires appeared to be ultimately stochastic throughout the analysis. The design of arbitrary categories to rank incidents objectively was deemed the best way to compare events and meet the goal of developing a methodology capable of forecasting the likelihood of extreme fire behavior.

**Table 2.** Categorical scores for percentiles of ERC, RH, and wind speed.

| ERC Percentile | RH Percentile | Wind Percentile | Score |
|:---:|:---:|:---:|:---:|
| 0–50th | 100–51st | | 0 |
| 50–59th | 50–41st | 0–20th | 1 |
| 60–69th | 40–31st | 20–40th | 2 |
| 70–79th | 30–21st | 40–60th | 3 |
| 80–89th | 20–11th | 60–80th | 4 |
| >90th | <11th | >80th | 5 |

### 2.3. Comparison to Observational Data

To test the reliability of the scoring methodology using observational data, a geographically diverse sample of 55 remote automatic weather stations (RAWS) were selected from the Western Regional Climate Center RAWS USA Climate Archive (https://raws.dri.edu accessed on 18 March 2023). Hourly observations were collected for the period of record. The maximum observed wind speed, gust speed, and minimum observed RH were extracted for each 24-hr period. RAWS wind is measured at 20-feet (6-m). RAWS gust speed is the instantaneous speed of the wind during that hour, while the standard wind speed measurement is a ten-minute average [42]. Maximum gust speed measurements were preferred over wind speed after it was observed in both gridded and hourly observations that wind speeds rarely approached the criterion value listed for each FWZ. RFW wind criteria exceeded the 90th percentile more than 95% of the time in both gridded and hourly datasets. For gust observations, however, RFW criteria had a median percentile value of 68. Gust observations also approached or exceeded criteria values more frequently, suggesting that gust speed measurements may have been the original point of reference when existing criteria were established. Gust observations were quality controlled by excluding values greater than three times the 80th percentile of all station measurements for the period of record [43]. By selecting the maximum hourly gust observation for each 24-hr period, we negated any potential weakness from using gridded wind speeds calculated at 1300 LST. Existing RFW warning criteria for each FWZ were then transformed into percentiles using RAWS gust speed and RH observations and scored via the categories listed in Table 2. As mentioned above, no standardized integration of fuels information is currently utilized for RFWs, therefore ERC observations from RAWS were unable to be compared to an existing criterion. The recommended inclusion of ERC percentiles in RFW criteria is intended to adequately address the fuels component.

### 2.4. Application to Previously Issued Warnings

To further validate the use of percentile scores, a dataset of all 2020 RFWs issued in a geographically diverse sample of 20 states were collected from the Iowa State University Iowa Environmental Mesonet archive of NWS warning products (https://mesonet.agron. iastate.edu/vtec/search.php#byugc/MTZ067 accessed on 18 March 2023) (see Table 1). A total of 5195 RFWs were retrieved. For RFWs that spanned more than two days, the median day was selected. For RFWs spanning only two days, the date before the expiration date was used if the warning expired before 1300 LST. If the warning expired after 1300 LST, the expiration date was used. This was done to capture the exact date accompanying optimal burning conditions for which the warning was likely issued. To obtain meteorological

information corresponding to each warning, a geographical reference location for each FWZ was chosen (in order of preference) from a RAWS location or the location of a previously documented large fire. RAWS locations are intended to be representative of the entire FWZ [42]. RFWs were not tested if they occurred in a FWZ where neither RAWS nor large fire information was available. A final sample of 3622 RFW locations was extracted from gridded values of ERC, RH, and wind speed datasets spanning 1999–2014 to match the large fire sample period. Due to the size and scope of this analysis, 1300 LST gridded wind speeds were utilized as a substitute for gust speeds despite their relatively weak correlation to observations [39] (p. 33). Since the percentiles calculated are relative only to other gridded values, the local severity of wind speed values was likely preserved. Relative proportionality between wind speed and gust speed has been previously demonstrated [44]. ERC, RH, and wind speed values corresponding to each RFW then were converted to percentiles and a score assigned to each.

## 3. Results

### 3.1. Individual Parameter Scores from Large Fire Occurrence

　　ERC, RH, and wind speed scores of 4 or greater (refer to Table 2) consisted of 69%, 62%, and 51%, respectively, of all large fire occurrences where the maximum rate of spread exceeded statewide breakpoints. These results are similar to Clark [10], where RFWs coinciding with ERC values above the 90th percentile exhibited superior skill in predicting large fire days, as well as work done by Lindley [45] where 2-m relative humidity and 20-foot (6-m) wind speed met RFW criteria in 64% of large fires on the southern great plains from 2006–2010.

　　Figure 3 demonstrates a strong likelihood for larger rates of spread as the percentile increased (in the case of ERC) or decreased (in the case of RH). This example consists of large fires exceeding the statewide ROS breakpoint in Idaho; other states (not shown) demonstrate similar results. It is worth mentioning that this pattern is not linear; low rates of spread are still common in the presence of extreme conditions, perhaps a due to the stochastic nature of fire, but also potentially a result of human action such as aggressive initial attack response. While most large fires occurred during severe conditions, there were still outliers occurring at near average conditions even above breakpoint rates of spread.

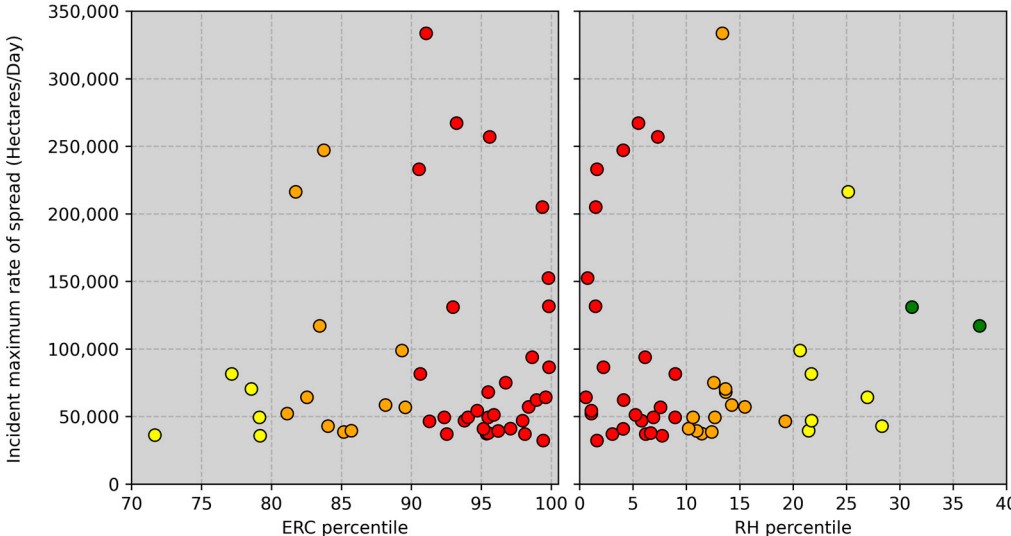

**Figure 3.** ERC and RH percentiles for large incidents exceeding the statewide breakpoint daily maximum ROS in ID, 1999–2014. Colors correspond to numerical scores in order, red being 5, orange 4, yellow 3, green 2, and blue 1 (refer to Table 2 for exact scoring criteria).

　　A relationship of wind speed percentiles with increasing rates of spread is visually apparent in states with predominant fuel types more susceptible to wind driven fire (e.g.,

grass fuel, Figure 4), and less apparent in states where timber constitutes the primary fuel type (Figure 5). Preliminary observation supports this assumption, as the tenth percentile wind speed for Oklahoma and Oregon are almost identical (0.94/1.02 m s −1; 2.1/2.3 mph), though wind speeds in the 0–20th percentile in Oregon depict a variety of rates of spread. Conversely, wind speeds in the 0–20th percentile in Oklahoma depict very low rates of spread. Above the ROS breakpoint wind percentiles Oklahoma and Oregon again appear to have almost identical wind speeds (2.8/3.0 m s −1: 6.2/6.7 mph) but drastically different rates of spread resulting from such. Rates of spread are also generally higher in Oregon, as the maximum daily rate of spread breakpoint is 49,500 hectares compared to 5000 hectares in Oklahoma.

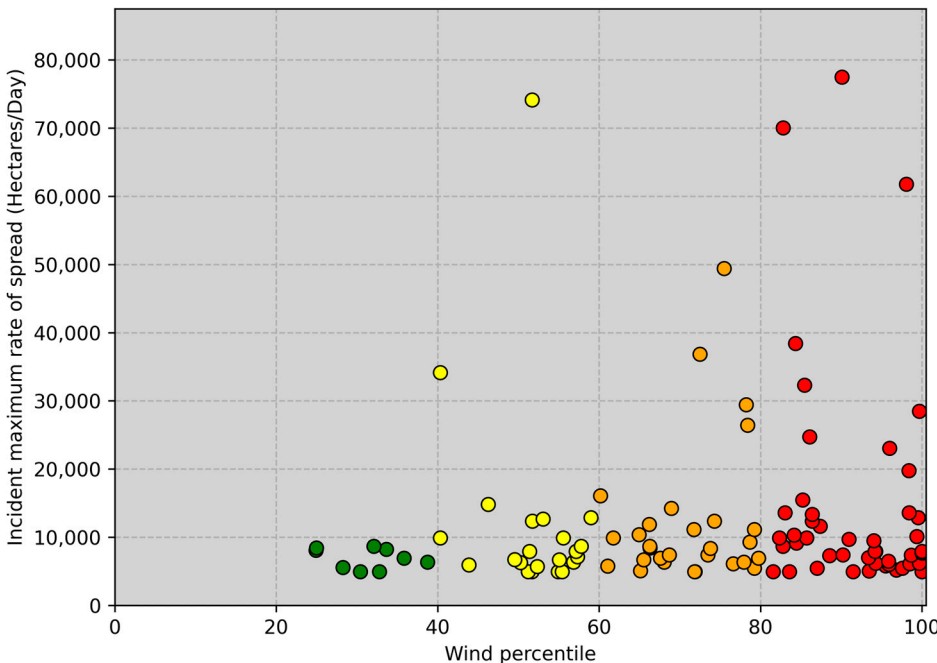

**Figure 4.** Wind speed percentiles for large incidents exceeding the statewide ROS breakpoint in OK, 1999–2014. Color scheme same as for Figure 3.

Additional analysis concerning the frequency of agreement between ERC and RH percentiles demonstrated that ERC and RH scores agreed 50% of the time throughout all sampled incidents, ERC and wind speed 20%, and wind speed and RH 23%. This is reasonable given the incorporation of RH into the calculation of ERC [46], though it is worth noting a lack of agreement 50% of the time, demonstrating independence between RH and ERC (i.e., cases with high ERC but low RH scores which still result in increased rates of spread and vice versa). Occasionally incidents occurred at very low scores for one or more variables (i.e., RH percentiles above 40, wind percentiles below 20, etc.). These outliers were not excluded from the analysis to highlight where fire behavior did not perfectly align with weather/fuel severity given the occasionally stochastic nature of wildfires. Notably, 72% of the time ERC and RH scores agreed both variable scores were 5, demonstrating strengthening alignment as conditions become more severe.

### 3.2. Total Scores from Large Fire Occurrence

Within the entire dataset (684 qualifying), 463 incidents (68%) obtained a total score of 11 or greater. The frequency of each score is shown in Figure 6. The tendency for higher scores as rate of spread percentile increases demonstrates the ability of the scoring methodology to accurately represent increasingly extreme fire behavior.

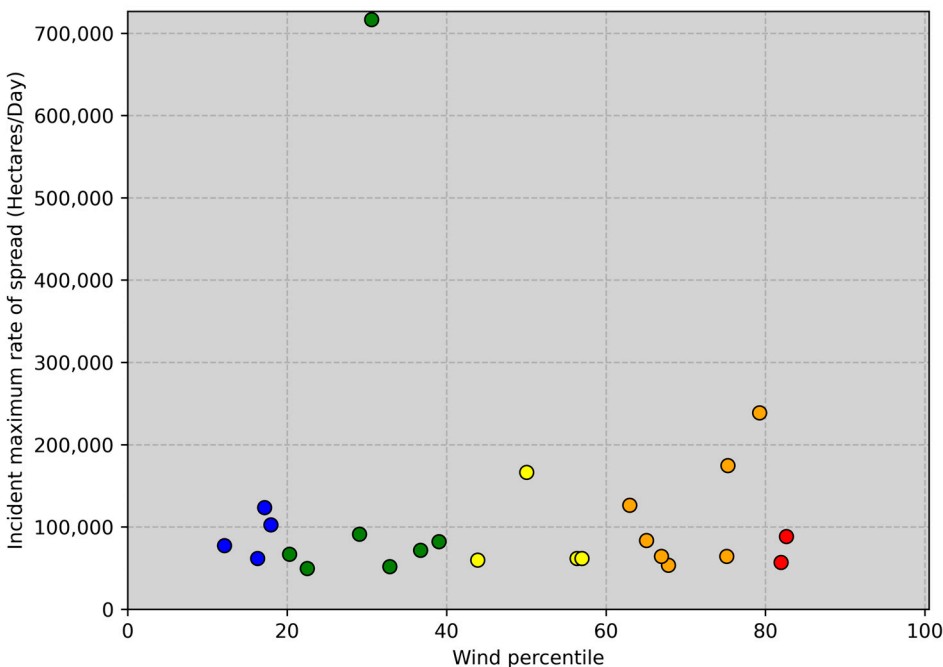

**Figure 5.** Wind speed percentiles for large incidents exceeding the statewide ROS breakpoint in OR, 1999–2014. Color scheme same as for Figure 3.

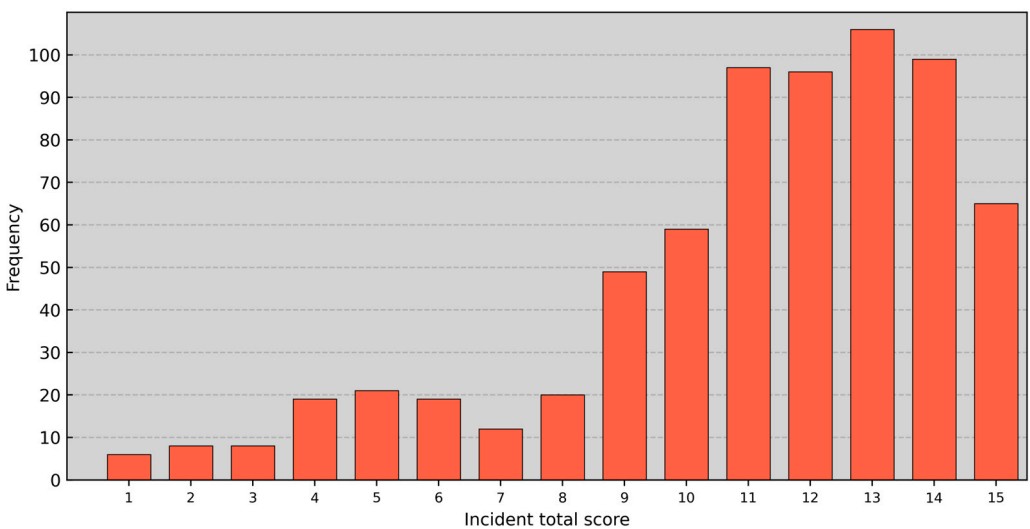

**Figure 6.** Frequency distributions of each total score for incidents exceeding statewide ROS breakpoint; all states 1999–2014.

### 3.3. Introduction of a Two-Step Decision Matrix

The resulting total scores are combined into a two-step decision matrix (Figure 7). Parameters may be individually scored and combined in any order (reference Table 2). Any combination is valid to achieve the same result. Here ERC and RH percentiles are scored together first, then compared to wind percentiles. Displaying the scoring methodology in matrix format provides a simplified reference tool for forecasters and easily identifies conditions warranting a RFW issuance. WFOs may adopt this format and replace each percentile category with corresponding actual values of ERC, windspeed, and RH, allowing for quick reference during daily operations. It is recommended that the threshold for considering a warning is placed at a score of 11, clearly corresponding to the increase in frequency of large fires with high rates of spread.

| | | RH Percentile | | | | |
|---|---|---|---|---|---|---|
| | | 50–41st | 40–31st | 30–21st | 20–11th | <11th |
| **ERC Percentile** | 50–59th | 2 | 3 | 4 | 5 | 6 |
| | 60–69th | 3 | 4 | 5 | 6 | 7 |
| | 70–79th | 4 | 5 | 6 | 7 | 8 |
| | 80–89th | 5 | 6 | 7 | 8 | 9 |
| | 90th+ | 6 | 7 | 8 | 9 | 10 |

| | | Combined ERC and RH Percentile Score | | | | | | | | |
|---|---|---|---|---|---|---|---|---|---|---|
| | | 2 | 3 | 4 | 5 | 6 | 7 | 8 | 9 | 10 |
| **Wind Percentile** | 0–20th | 3 | 4 | 5 | 6 | 7 | 8 | 9 | 10 | 11 |
| | 20–39th | 4 | 5 | 6 | 7 | 8 | 9 | 10 | 11 | 12 |
| | 40–59th | 5 | 6 | 7 | 8 | 9 | 10 | 11 | 12 | 13 |
| | 60–79th | 6 | 7 | 8 | 9 | 10 | 11 | 12 | 13 | 14 |
| | 80th+ | 7 | 8 | 9 | 10 | 11 | 12 | 13 | 14 | 15 |

**Figure 7.** Categorical scores for percentiles of ERC, RH, and wind speed as a two-step decision matrix. Yellow boxes correspond to scores that suggest a warning may be necessary; red boxes indicate more severe conditions.

*3.4. Current RFW Criteria Scores Using Observational Data*

Using hourly data gathered from RAWS stations, RH, and gust speed values meeting current RFW criteria were scored using the same method, resulting in an average score of 3 for RH criteria and 4 for wind criteria. This results in an average combined score of 7, and under worst case scenario ERC conditions (5) the threshold for a RFW would presently sit at total score of 12. Note that the preceding analysis of large fire incidents relied upon gridded data for wind speed; here RAWS hourly observations of gust speed are substituted for a more accurate comparison to actual criteria values. There was some regional variation in how criteria scored; for example, RFW criteria for RH hover around the 50th percentile in the Great Basin but are under the 10th percentile in New York. RFW criteria in Montana and Washington were below the 50th percentile gust speed according to the sampled stations (8 total). These results allude to the possibility that this new methodology may be more strict than existing RFW criteria, which could eliminate over forecasting (which has been a concern for some fire management agencies) while still allowing some amount of forecaster discretion given threshold scores.

RH observations from RAWS were generally higher than gridded values, indicating observed conditions may be less severe than implied by gridded data. Wind speed observations from RAWS mean wind velocity were also generally higher than gridded values, but substantially lower than gust speed measurements, indicating that by selecting gust speed, the analysis focused on the worst-case scenario. As previously mentioned, scoring RH and wind speed criteria against each other is a current operational practice of some WFOs. Where this was observed, relative agreement between existing criteria and our proposed methodology was commonly found. In cases where scores did not agree with existing criteria, the method described herein favored a slightly stricter approach.

*3.5. Scoring of Previous RFWs*

Testing this methodology with RFWs issued in sampled states throughout 2020 revealed congruent results (see Figure 8). ERC, RH, and wind scores of 4 or greater consisted of 85%, 78%, and 78%, respectively, of all situations where RFWs were issued. In 83% of instances, the score of a RFW was greater than or equal to 11, implying that this method would agree with a large majority of current RFW criteria.

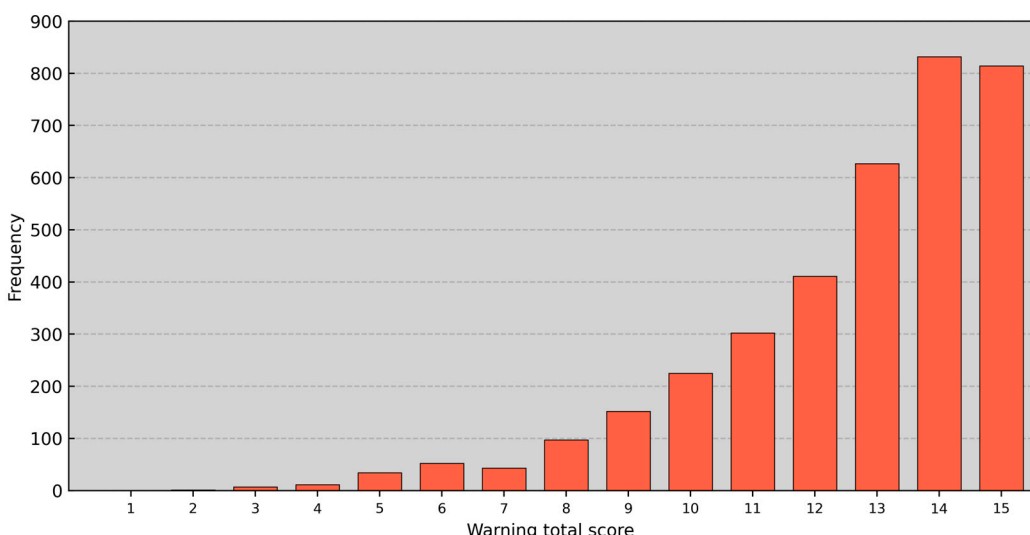

**Figure 8.** Frequency of 2020 RFW total scores for all states.

## 4. Discussion

### 4.1. Expanding Current Warning Messaging

Currently, the NWS may issue FWWs prior to issuing RWFs, the latter is often a chronological progression of the former; however, forthcoming work done by co-author T. Wall demonstrates that fire management does not always successfully distinguish between the two products and instead views FWWs as less severe versions of RFWs. This presents an opportunity to replace the current FWW/RFW products with a two-tiered warning system that is better able to communicate severity information to fire managers. Further assessment is needed to determine if changes in FWW and RFW terminology are warranted. This would be especially prudent to consider as part of any matrix integration into operations. Conditions escalating to a score of 13, 14, or 15 should be considered as the more extreme conditions, and therefore a second category of warning with different language may be appropriate. It would be appropriate to communicate the historical severity of RFWs when approaching extreme percentiles, such as for issuance of a particularly dangerous situation. It may also be helpful to consider issuing RFWs for larger scale meteorological events rather than specific windows of time. Though these conditions are obviously peaking in severity for limited amounts of time and the understanding and perception from fire management may be more actionable if more detailed meteorological background conditions are cited in warning messaging. There is also evidence to suggest that the amount of time that warning conditions persist has little impact on the likelihood of fire ignition [45], implying that RFW criteria with a minimum duration may not be necessary.

### 4.2. The Inclusion of Uncertainty Information

The literature surrounding severe weather warnings in general demonstrates a substantial benefit from including uncertainty information, especially when forecasts surround high impact events or are targeted at sophisticated user groups [47,48]. Emergency managers often understand forecast uncertainty [49], and directly benefit from added qualitative information [50]. Brown and Murphy [51] specifically addressed fire weather forecasts, concluding that uncertainty information was necessary for fire managers to make rational decisions. It is therefore reasonable to assume RFWs could be improved by the addition of uncertainty information as well as the communication of the relative severity of the warning. Our proposed method would allow for RFWs to be comparatively ranked and severity easily identified, as well as mitigate any opportunities for heuristics or experience to impact the forecast [52].

*4.3. Limitations*

While the inclusion of a standardized fuel dryness criteria is necessary, some practitioners may be concerned that ERC-G(Y) is not representative of local conditions and, thus, cannot adequately proxy all fuel types. However, as noted earlier, ERC-G(Y) is a commonly used indicator for many applications. The use of percentiles overcomes the large variation of ERC magnitude in relation to fire occurrence across the country. This analysis utilized ERC percentiles derived from gridded data standardized to fuel model G (as per convention).

There are known limitations to the accuracy of gridded data with respect to its ability to capture micro-climates [39]. This may have impacted the overall scores of large fires presented herein; however, preliminary assessment with RAWS observations suggests that this methodology would result in less RFWs than initially suspected with use of gridded data, as most existing criteria scored higher within the gridded dataset used here. This could shift concern away from false positives but may require local WFO adjustments in warning thresholds if extreme fire behavior events are consistently being overlooked. Limited work was performed regarding the statistics of overall score occurrence outside of large fire incidents or RFW issuance, and while comparisons to current criteria do not appear to imply continued or increased over forecasting, no definitive frequency analysis currently demonstrates otherwise.

## 5. Conclusions

RFWs are a valuable tool for fire management. The ability to accurately forecast the potential occurrence of extreme fire behavior will expose fire practitioners to less risk [35], allow for better strategic planning and resource allocation by fire management [37], and potentially affect decisions heavily impacting the public [53]. Current RFW criteria are nationally inconsistent and often lack a formal integration of fuels information. These issues pose potential problems with issuance consistency and relevancy, as well as limiting forecasters' ability to accurately judge historical severity. Here, we have developed a new methodology for RFW criteria based off documented occurrences of extreme fire behavior, specifically rates of spread above the statewide 80th percentile, as available from the ICS-209 database. Percentile analysis of each incident was made possible by the availability of a high-resolution gridded surface climatology. Categories of ERC, RH, and wind speed were assigned a score of 1 to 5, and the additive combination of all three scores provided for a final numerical score ranging from 0 to 15. This score given in a decision matrix form indicates the extremity of conditions for issuing a RFW and FWW.

Care was taken to gauge how current RFW criteria would score in the matrix from a select sample of FWZs by extracting and scoring RFW criteria percentiles from hourly RAWS observations. Several forecasting matrices currently being used by WFOs were also examined by comparing RFW criteria to their corresponding historical percentile. Though more work is needed to examine the frequency and distribution of RFWs under this new method in relation to current operational forecasts, it is not expected to be radically different from present issuance and may err on the strict side of some WFOs forecasting practices.

Our project introduces a methodology for RFW issuance guidance for potential use by the National Weather Service. As such, quantitative verification was not performed such as that carried out by Clark et al. [10] as this would be more of a specific forecast issuance exercise. However, it is of interest to compare this new methodology with current operations. RFWs issued in 2020 were analyzed using the same method, demonstrating a marginal agreement of current criteria with the new proposed method despite a lack of national standardization. This suggests that the application of this method would cause minimum disruption to current forecasting practices once the details of integrating historical percentiles into operations were resolved. By utilizing the proposed methodology, forecasters would benefit from the ability to compare historical percentiles and more accurately judge severity, as well as being able to communicate this information to fire managers.

### 5.1. Summary of Results

1.  A standardized methodology was used to create a two-step decision matrix that condenses meteorological information into numerical thresholds demonstrated to correspond with higher likelihoods of extreme fire behavior.
2.  ERC, RH, and wind speed scores of 4 or greater consisted of 69%, 62%, and 51%, respectively, of all large fire occurrences where the maximum rate of spread exceeded the statewide breakpoint.
3.  More than two thirds (68%) of large fires produced a matrix score of 11 or higher.
4.  A total of 83% of RFWs issued in 2020 produced a matrix score of 11 or higher.
5.  As rate of spread increases, the likelihood of a higher matrix score also increases.

### 5.2. Summary of Recommendations

1.  NWS should nationally utilize a decision matrix approach for determining a RFW and FWW.
2.  Energy release component from NFDRS should be utilized as part of RFW criteria.
3.  NWS WFOs should base RFW criteria on historical percentiles of ERC, RH, and wind gust. These percentiles should be scored according to the categories depicted in Table 2. Days exceeding a score of $\geq 11$ should be considered as having elevated potential for extreme fire behavior.
4.  Numerical percentile values for these three elements would have to be calculated from historical data for FWZs and made available in the operational environment. However, similar with other NWS watches/warnings, RWW/RFW should be issued based on polygon type boundaries versus FWZs.
5.  In the context of hazard simplification, it is recommended to continue use of a two-tiered (watch and warning with special provision for PDS) set of fire weather products for clarity.
6.  Descriptive information regarding historical and seasonal severity should be included with each FWW/RFW.
7.  WFOs should frequently and freely communicate with fire management in order to ascertain forecast effectiveness and discuss potential impacts of the forecast.
8.  This analysis focused exclusively on fire management as the intended audience. Since FWW/RFWs are also being used by the public in some cases, further work is necessary to determine appropriate messaging for public use.

**Author Contributions:** Conceptualization, S.J. and T.B.; methodology, S.J.; software, S.J.; validation, S.J.; formal analysis, S.J.; investigation, S.J., T.B. and T.W.; resources, S.J., T.B. and T.W.; data curation, S.J.; writing—original draft preparation, S.J.; writing—review and editing, T.B. and T.W.; visualization, T.B. and T.W.; supervision, T.B. and T.W.; project administration, T.B. and T.W.; funding acquisition, T.B. All authors have read and agreed to the published version of the manuscript.

**Funding:** This research was funded by NOAA NWS CSTAR grant number NA19NWS4680002 and NOAA RISA CNAP grant number NA17OAR4310284.

**Data Availability Statement:** The ICS 209 dataset of large fires occurring in the United States 1999–2014 can be found at https://data.nal.usda.gov/dataset/data-all-hazards-dataset-mined-us-national-incident-management-system-1999--2014 accessed on 18 March 2023. All 4-km gridded surface climatology datasets may be found at https://www.climatologylab.org/gridmet.html accessed on 18 March 2023. RFWs issued in 2020 were collected from the Iowa State University Iowa Environmental Mesonet archive of NWS warning products https://mesonet.agron.iastate.edu/vtec/search.php#byugc/MTZ067 accessed on 18 March 2023. Data from remote automated weather stations (RAWS) was selected from the Western Regional Climate Center RAWS USA Climate Archive (https://raws.dri.edu accessed on 18 March 2023). An example of an agency fire danger operating plan may be found here: https://gacc.nifc.gov/nwcc/districts/CCCC/nfdrs2016/docs/WW_FDOP_final_draft_v20200428.pdf accessed on 18 March 2023.

**Acknowledgments:** The authors thank the National Weather Service for their contributions and support throughout the project. Project collaborators included Donald Dumont, Robyn Heffernan, Heath Hockenberry, Matt Jolly, Phillip Manuel, Chuck Redman, and Larry VanBussum. A special thanks to Todd Lindley and Nick Nauslar for their continued involvement and suggestions.

**Conflicts of Interest:** The authors declare no conflict of interest.

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
