# Peer review of "Development of a Decision Matrix for National Weather Service Red Flag Warnings"

_fire, doi:10.3390/fire6040168_

Round 1
Reviewer 1 Report
This is a commendable paper. It is characterized by clarity and good timing. I would, however, suggest three potential improvements in the overall narration of the authors:
1. It would benefit the readership to receive additional information on the methodology of calculating the inflection point at which spread rates start rapidly increasing (L. 154-157). Although appearing a "mathematistry" issue, it seems to be of primary importance for the entire research.
2. Rescaling x-axis Fig. 3, ERC and RH percentiles, by pruning non-responsive bins, would increase the resolution of and information on the phenomenon. One can see this in the proposed Figure 7.
3. Perhaps, one line of explanation why extreme values are included in, e.g., Fig. 5, and not considered as unique outliers would clarify this issue.
Author Response
We greatly appreciate the Reviewer 1 comments and have integrated the following revisions into our manuscript:
1. It would benefit the readership to receive additional information on the methodology of calculating the inflection point at which spread rates start rapidly increasing (L. 154-157). Although appearing a "mathematistry" issue, it seems to be of primary importance for the entire research.
We have included a more detailed statement on the procedure used to calculate inflection points (lines 157-159). “These breakpoints were obtained using an analytical method developed in Python specifically designed to find the knee or elbow of curves by calculating the slope of a line drawn from the minimum to maximum value, rotating the data so that the slope of this line is zero, and then finding the minimum value of the rotated data, corresponding the inflection point of the curve [38].”
2. Rescaling x-axis Fig. 3, ERC and RH percentiles, by pruning non-responsive bins, would increase the resolution of and information on the phenomenon. One can see this in the proposed Figure 7.
The x-axis of Fig. 3 has been rescaled to eliminate sections of empty space and provide a more concise visual aid.
3. Perhaps, one line of explanation why extreme values are included in, e.g., Fig. 5, and not considered as unique outliers would clarify this issue.
In reference to Fig. 5, valid outliers were not removed from the analysis, however, and a sentence was added to clarify our rational (lines 300-304). “Occasionally incidents occurred at very low scores for one or more variables (i.e., RH percentiles above 40, wind percentiles below 20, etc.). These outliers were not excluded from the analysis to highlight where fire behavior did not perfectly align with weather/fuel severity given the occasionally stochastic nature of wildfires.”
We hope these revisions thoroughly address the comments and improve the quality of the manuscript.
Reviewer 2 Report
General Comments:
The manuscript is organized, well-written, and introduces a novel concept. It’s an extremely interesting read overall!
The introduction provides a thorough background on the many challenges and considerations into which components should go into the proposed RFW criteria. The methodology is clearly explained and thought-out. The matrices are very intriguing and could prove beneficial to the WFOs and decision makers. The discussion and conclusions connect the methods back to current literature and provide a good summary of the findings and potential implications of this proposed method.
Specific Comments:
· - Table 1 – How did you sample the RFWs (right-most column)?
· - Lines 182-184 – Is there an article you can cite to back up this sentence?
· - Line 350: Change “winds peed” to “wind speed”.
Author Response
We greatly appreciate the Reviewer 2 comments and have integrated the following revisions into our manuscript:
1. Table 1 – How did you sample the RFWs (right-most column)?
Clarifying language was added around the sampling methodology of RFWs (line 236), specifically the word ‘all’ to indicate that all RFW in the listed states were initially collected and then further vetted according to the methods described. “To further validate the use of percentile scores, a dataset of all 2020 RFWs issued in a geographically diverse sample of 20 states were collected from the Iowa State University Iowa Environmental Mesonet archive of NWS warning products (https://mesonet.agron.iastate.edu/vtec/search.php#byugc/MTZ067 ) (see Table 1).”
2. Lines 182-184 – Is there an article you can cite to back up this sentence?
Lines 182-184 (now lines 185-186) were rephrased to imply that this was simply an observation from our dataset, not necessarily a previously known scientific phenomenon, thus there is no specific literature to cite.
3. Line 350: Change “winds peed” to “wind speed”.
Changed line 358 to “wind speed”.
We hope these revisions thoroughly address the comments and improve the quality of the manuscript.
Reviewer 3 Report
The article is devoted to an undoubtedly topical problem: predicting wildfires. A new forecasting method using a wide set of historical data on fires in the U.S. states and mathematical processing of the data is proposed. New parameters were introduced and the model was verified taking into account the fires that occurred and their characteristics. The topic and abstract correspond to the article. Goals and objectives are set, and scientific novelty is defined. The literature review is at a good level, there are references to previous works, which do not contradict this study. The results obtained correspond to the objectives. The article is neatly done. It is not clear only how the fire department will directly use the results. It will be necessary to make a more engineering interpretation of the results.
Author Response
The article is devoted to an undoubtedly topical problem: predicting wildfires. A new forecasting method using a wide set of historical data on fires in the U.S. states and mathematical processing of the data is proposed. New parameters were introduced and the model was verified taking into account the fires that occurred and their characteristics. The topic and abstract correspond to the article. Goals and objectives are set, and scientific novelty is defined. The literature review is at a good level, there are references to previous works, which do not contradict this study. The results obtained correspond to the objectives. The article is neatly done. It is not clear only how the fire department will directly use the results. It will be necessary to make a more engineering interpretation of the results.
We added some text to line 454 to clarify that the product is aimed for use by the National Weather Service. They will provide the necessary interpretation for fire department usage.
Our project introduces a methodology for RFW issuance guidance for potential use by the National Weather Service.